# Soluble RAGE enhances muscle regeneration after cryoinjury in aged and diseased mice

Naftali Horwitz[1,2,3¤]*, Michael Florea[1,2,4], K. C. Medha[1,2], Tina Liu[1,2], Vivian Garcia[1,2,4], Rebekah Kim[1,2,5], Amy Lam[1,2], Kathleen Messemer[1,2], Christopher Rios[1,2], Albert E. Almada[1,2,6] Amy J. Wagers[1,2,7,8]*

1 Harvard Stem Cell Institute, Harvard University, Cambridge, Massachusetts, United States of America, 2 Department of Stem Cell and Regenerative Biology, Harvard University, Cambridge, Massachusetts, United States of America, 3 Harvard Chemical Biology Ph.D. Program, Harvard University, Cambridge, Massachusetts, United States of America, 4 Harvard Ph.D. Program in Biological and Biomedical Sciences, Division of Medical Sciences, Harvard University, Boston, Massachusetts, United States of America, 5 Harvard Immunology, Division of Medical Sciences, Harvard Medical School, Boston, Massachusetts, United States of America, 6 Department of Orthopaedic Surgery and Department of Stem Cell Biology and Regenerative Medicine (SCRM), University of Southern California, Los Angeles, California, United States of America, 7 Paul F. Glenn Center for the Biology of Aging, Harvard Medical School, Boston, Massachusetts, United States of America, 8 Section on Islet Cell and Regenerative Biology, Joslin Diabetes Center, Boston, Massachusetts, United States of America

¤ Current Address: Department of Obstetrics & Gynecology at Columbia University Irving Medical Center, New York, NY, 10032, USA

* nh2395@cumc.columbia.edu (NH); amy_wagers@harvard.edu (AJW)

## Abstract

The Receptor for Advanced Glycation End Products (RAGE), classically considered a mediator of acute and chronic inflammatory responses, has recently been implicated by genetic knockout studies as a regulator of skeletal muscle physiology during development and following acute injury. Yet, the role of its soluble isoform, soluble RAGE (sRAGE), in muscle regeneration remains relatively unexplored. To address this knowledge gap, Adeno-Associated Virus (AAV) mediated and genetic knockin supplementation strategies were developed to specifically assess the effects of changing levels of sRAGE on muscle regeneration. We evaluated general muscle physiology and histology, including central nucleation, and myofiber size. We found that acute induction of sRAGE in aged and atherosclerotic animals accelerates muscle repair after cryoinjury. Similarly, genetic modification of the endogenous *Ager* gene locus to favor production of sRAGE over transmembrane RAGE accelerates repair of cryo-damaged skeletal muscle. However, increasing sRAGE via AAV delivery or using our transgenic mouse lines had no impact on muscle repair in aged or diseased mice after barium chloride (BaCl$_2$) injury. Together, these studies identify a unique muscle regulatory activity of sRAGE that is variable across injury models and may be targeted in a context-specific manner to alter the skeletal muscle microenvironment and boost muscle regenerative output.

## Introduction

Skeletal muscle is a highly regenerative tissue whose recovery after injury depends on the coordinated activity of an adult muscle stem cell (MuSC) population. While the reparative

**Data availability statement:** The datasets and materials used and analyzed during the current study are available from Dryad repository under accession DOI: 10.5061/dryad.3j9kd51vx.

**Funding:** This study was supported by a training grant award from the National Institute of Diabetes and Digestive and Kidney Diseases (2T32DK007260) to NH, awards from the Paul F. Glenn Medical Foundation and the National Institutes of Health (NIH) (R01AG048917) to AJW, and awards from the Burroughs Wellcome Fund, NIH diversity supplement (3R01AG048917-02S1), and NIH (1R01AR080753) to AEA. The funders had no role in study design, data collection and analysis, decision to publish, or preparation of the manuscript.

**Competing interests:** A.J.W. is a scientific advisor for Kate Therapeutics and Frequency Therapeutics, as well as a co-founder and scientific advisory board member and holds private equity in Elevian, Inc., a company that aims to develop medicines to restore regenerative capacity. Elevian also has provided sponsored research to the Wagers lab.

response of MuSCs is typically robust early in life, our bodies muscle repair ability tends to decline with age and in patients suffering from age-associated tissue dysfunction like atherosclerosis and diabetes [1–4]. These conditions are characterized by reduced MuSC activity, delayed repair after injury, and chronic low-grade inflammation in the muscular system.

The Receptor for Advanced Glycation End Products (RAGE) is a multiligand mammalian transmembrane receptor whose signaling cascades directly influence key cellular functions like migration, proliferation and apoptosis [5,6]. Abnormal regulation of the receptor and its associated RAGE ligands result in the development of several chronic pathologies like diabetes, cardiovascular disease and cancer [7–9]. In skeletal muscle specifically, RAGE acts as a central mediator of muscle development and homeostasis as well as a major driver of repair after acute muscle injury [10–12]. Conversely, in the context of aging, RAGE may also contribute to muscle damage and atrophy but its mechanism is less clear [13–15]. Therefore, RAGE and its downstream signaling networks need to be tightly controlled to prevent its overactivity, which leads to negative effects on the repair and maintenance of tissues as we age.

Interestingly, the soluble isoform of RAGE (sRAGE) retains all the ligand binding domains found in RAGE except that it lacks a transmembrane domain, leading to its release into the outside micro-environment. Previous groups have shown that sRAGE essentially functions as a soluble decoy receptor by blocking the negative effects of RAGE signaling and delaying pathology in several disease mouse models [16–19]. More recently, sRAGE has been shown to be expressed in the blood of long-lived centenarians, and thus, some have proposed its role as a human 'Longevity' molecule [20–23]. This initial interest has led to the development of several RAGE-blocking molecules that are currently being tested in humans in the clinic [24–26]. However, the role of sRAGE in skeletal muscle function and repair after injury remain relatively unexplored.

In this study, we report that sRAGE supplementation accelerates skeletal muscle repair after cryo-injury but not after barium chloride injury. We validated sRAGE's pro-regenerative effects using an Adeno-Associated Virus (AAV) sRAGE supplementation strategy in aged and atherosclerotic animal models, as well as an endogenous, life-long, transgenic sRAGE mouse model. Overall, this work highlights the complex and injury-specific role of sRAGE in modulating muscle regenerative function in healthy, aged, and diseased vertebrate animal models.

## Materials and methods

### Generation of *AGER* strains

The Harvard University Institutional Animal Care and Use Committee (IACUC) approved all animal protocols used in this study (IACUC #29-14). Homozygous *AGER^{s/s}* (C57BL/6J) mice [27] were obtained from L. Lin and K. Perdue at the National Institute of Aging (NIA) and crossed to C57BL/6J (Jackson Labs) mice to generate *AGER^{s/+}* animals. Heterozygous *AGER^{+/-}* (C57BL/6J) [28] were obtained from C. Lee at Brown University and crossed to generate *AGER^{-/-}* animals. Resulting animals were crossed with *AGER^{s/s}* animals to generate AGER^{s/-} animals. Cohorts of *AGER^{s/s}*, *AGER^{s/+}*, *AGER^{s/-}*, *AGER^{-/-}*, *AGER^{+/-}*, and *AGER^{+/+}* used for a given experiment regularly consisted of mice from multiple litters. Adult mice were age-matched (no more than 3 weeks differences in ages within the 7–8 months age range) and sex-matched for each experiment. Only male animals were included.

### Additional mouse strains

*C57BL/6J, C57BL/10ScSn-Dmd^{mdx}/J, B6.129P2-Apoe^{tm1Unc}/J, and B6.BKS(D)-Lepr^{db}/J* mice were handled, housed, and matched by age and sex as outlined above. Animals from multiple litters were often used in each experiment. Male mice (ranging from 2–20 months of age) were included in this study.

## Histology and processing of skeletal muscle samples

Tibialis Anterior (TA) muscles were collected from mice, immersed in O.C.T., and frozen in 2-methylbutane chilled with liquid nitrogen for ~30 seconds. Frozen TAs were placed at −80°C for at least two days. TAs were then cryo-sectioned at 10–15 depths covering the middle two-thirds of the muscle with a 12 μm thickness per section. Resulting sections were used for immunofluorescence staining (IF) for DYSTROPHIN detection or for Hematoxylin and Eosin (H&E) staining to assess the cross-sectional area (CSA) of regenerating (centrally-nucleated) muscle fibers.

## In vivo regeneration assays

Barium Chloride (BaCl2) injury for *in vivo* regeneration analyses was done by injecting 50 μL of 1.2% BaCl2 (Sigma-Aldrich) into the Tibialis Anterior (TA) muscle of animals. Cryoinjuries were conducted by making a small skin incision, exposing the TA muscle, and pressing a cold (cooled in dry-ice) flat metal rod to the muscle for 10 seconds. After 7 days, mice were euthanized, and regenerating TAs were collected and sectioned at 10–15 depths covering the middle two-thirds of the muscle with a 12 μm thickness per section. Slides were coded so that all subsequent analysis was blinded. Slides were stained with H&E and for each slide the section showing the most centrally nucleated fibers nearest to the injury was used for mean Cross Sectional Area (CSA) quantification. Four representative images (20X magnification) adjacent to the largest areas of injury within each section were taken for analysis. CSA of regenerating fibers was manually quantified using ImageJ (Fiji). Each sample was unblinded after analysis was finished and imported in Prism Graphpad.

## Satellite cell isolation by FACS

Muscles were isolated from animals and physically triturated with surgical scissors before being placed in enzyme mix consisting of 0.2% Collagenase type II (285 U/mg, Gibco #17101015) and 0.05% Dispase (1.81 U/mg, Gibco #17105041) in Dulbecco's Modified Eagle Medium (DMEM) for 15 minutes in a 37°C shaking incubator. Muscle was further triturated with a 5 mL pipet (pre-wet in FBS), until the homogenate smoothly moved through the pipette, and then placed back in the shaking incubator for 10 minutes. Enzyme activity was inactivated by adding Fetal Bovine Serum (FBS), followed by two consecutive washes with Phosphate Buffered Saline (PBS) spun at ~1600 rpm. Cells were filtered through a 70-micron cell strainer and spun at ~1260 rpm. Cells were then resuspended in staining media (2% FBS in Hanks Balanced Salt Solution (HBSS)) and incubated on ice for 30 minutes with fluorescently-conjugated antibodies to detect the following antigens: anti-CD31-APC (1:200, Clone 390 BioLegend), anti-Ly6A/E-APC (Sca-1) (1:200, Clone E3-161.7 BioLegend), anti-CD11b-APC (Mac-1) (1:200, Clone M1/70 BioLegend), anti-CD45-APC (1:200, Clone 30-F11 BioLegend), anti-Ter119-APC (1:200, Clone TER-119 BioLegend), anti-CD29-APC-Cy7 (β1-Integrin) (1:200, Clone HMβ1-1 BioLegend), anti-CD184- Biotin(CXCR4) (1:100, Clone 2B11 BioLegend). A secondary antibody, Streptavidin-PeCy7 (1:200, BioLegend) for CXCR4 detection, was incubated with the appropriate samples on ice for 20 minutes. Additionally, single color controls and Fluorescent minus-one controls (FMOs) were prepared for all experiments. Following antibody incubation, cells were washed twice with staining media, resuspended, and analyzed by FACS using a BD FACSAria III cell sorter and gated sequentially on physical parameters and live/dead markers (Propidium Iodide-negative/Calcein Blue-positive). Satellite cells were defined as the CD11b-; CD45-; CD31-; Ter119-; Sca1-; CD29 + (β1-Integrin); CD184 + (CXCR4) cell population as this gating strategy isolates a population of satellite cells which overlaps significantly (>90%) with other established positive selecting antigens (i.e., VCAM1 and α7-integrin/CD34) [29]. This

isolation process - from mouse muscle to FACS-ready mononuclear cells - typically requires 3–4 hours. Given our desire for a pure cell population, an additional 2–4 hours are required at the cell sorter as we typically double-sort our satellite cell population.

## Immunofluorescence (IF)

For DYSTROPHIN staining, TA muscle sections were thawed and incubated with blocking buffer (3% Bovine Serum Albumin (BSA), 5% Normal goat serum (NGS), and 0.1% Triton-X in PBS) for 1 hour at RT. Without washing, a solution of primary mouse anti-Dystrophin (1:100; Sigma D8168), and rabbit anti-laminin (1:200; Sigma L9393), in 3% BSA and 5% NGS was applied and incubated with sections overnight at 4°C. The next day, sections were rinsed 4x15 minutes in PBST and a mixture of secondary antibody containing goat anti-mouse IgG Alexa Fluor 555 (1:250, Invitrogen #A28180) and goat anti-rabbit IgG Alexa Fluor 488 (1:250, Invitrogen #A-11008) in 3% BSA and PBST was added for 1 hour at RT. Sections were then rinsed for 4x15 minutes in PBST. Sections were stained with Hoescht 33342 (1:10,000, Thermo Fisher #62249) and rinsed in PBST 3x5 min. Following washes, slides were mounted in VECTASHIELD Hardset Antifade Mounting Medium (Vector Laboratories H-1400-10).

## Cell transplantation

24 hours before transplantation, recipient *C57BL/10ScSn-Dmd^mdx^/J* mice were pre-injured by injecting 25 µL of 1.2% BaCl2 into the TA muscle. The following day, 3000 or 6000 donor fresh SCs from *AGER^s/s^*, *AGER^s/+^*, *AGER^s/-^*, *AGER^-/-^*, *AGER^+/-^*, and *AGER^+/+^* mice were isolated via FACS, re-suspended in 20 µL of staining media (2% FBS in HBSS) and injected into the pre-injured TAs of recipient mice. Three weeks post-injection, TAs were isolated, frozen and sectioned at 10–15 depths covering the middle two-thirds of the muscle with a 12 µm thickness per section. Sections were stained for DYSTROPHIN (as outlined in staining protocol above) and 3 depths of the whole muscle were imaged per sample on a Zeiss 880 Inverted Microscope. Data points represent the total number of Dystrophin + fibers that were averaged from at least 3 depths of the TA muscle.

## Western blot

Tissues were isolated and homogenized in RIPA buffer mixed with HALT phosphatase and protease cocktail mix using a gentleMACS tissue dissociator. The mixture was allowed to rest on ice for 20 minutes and homogenized again. Tissue lysate was spun down at 12,000 rpm at 4°C for 10 minutes and supernatant was collected. Leftover tissue lysate was stored at −80°C. Protein concentration was quantified using a Pierce Rapid Gold BCA Protein Assay Kit. Lysate was mixed with equal volume of sample buffer consisting of 95% 2X Laemmli Buffer and 5% beta mercaptoethanol and a total of 10 µg was run per lane on an AnykD Criterion TGX precast gel. Proteins were transferred using Trans-Blot Turbo PVDF transfer packs and membranes were incubated in 5% milk in TBST for 1 hour at RT. Membranes were then incubated overnight at 4°C with primary antibody (1:200, rabbit anti-RAGE, ab3611) in 5% milk. The next morning, membranes were washed 4x5 minutes in TBST and incubated in (HRP)-conjugated goat anti-rabbit IgG secondary antibody (1:500, #31460, Invitrogen) for 1 hour at RT. Membranes were washed 4x5 minutes in TBST and imaged on a BioRad GelDoc XR + and ChemiDoc XRS + System. For size reference, Precision Plus Protein™ Kaleidoscope™ Prestained Protein Standards were run alongside samples. Concomitantly, a separate gel was run with the same protein samples and incubated with a rabbit anti-GAPDH antibody (sc-32233, 1:200) to serve as a protein loading control.

## ELISA

Tissues were isolated and homogenized in RIPA buffer as described in the previous section. Otherwise, 50 μl of blood was collected from the tail vein of mice and mixed with 50 μl of RIPA buffer. Total Protein concentration was quantified using a Pierce Rapid Gold BCA Protein Assay Kit (Thermo Fisher #A55860). Mouse RAGE Quantikine ELISA (R&D Systems #MRG00) was performed according to the protocol provided by the manufacturer. In short, samples were diluted 1:10 in calibration diluent provided. Serial dilutions of the mouse RAGE standard were also prepared according to the manufacturer's instruction. Samples and standards were added to microplate strips provided (50uL) along with an equal amount of Assay diluent solution provided. The plate was allowed to incubate on an orbital shaker for 2 hours followed by an aspiration step and four repeated washes of the wells with wash buffer provided. Anti-RAGE polyclonal antibody conjugated to HRP was added to each well (100uL) and allowed to incubate on an orbital shaker for 2 hours followed by another aspiration step and four more washes. Substrate solution (100uL) was then added to each well and allowed to incubate, covered from light, for 30 minutes. Stop solution (100uL) was added and the plate was quickly analyzed on a microplate reader set to 450 nm. To correct for optical imperfections on the plate, the measurements were repeated at 540 nm and subtracted from the measurements at 450 nm.

## Blood glucose measurements

Blood was drawn from the tail vein and immediately sampled on a OneTouch® Ultra® 2 glucometer.

## Construction of AAV9-sRAGE

The scAAV-Cbh-sRAGE expression vector was cloned from an scAAV-CMV-GFP-bGHpA plasmid backbone, received as a generous gift from Vandenberghe lab at Harvard Med School/Mass General Brigham. The Cbh-miR122t-SV40pA expression construct was synthesized and cloned between the ITRs of the self-complementary backbone using NheI (NEB #R3131) and MscI (NEB #R0534) restriction enzyme sites. sRAGE coding sequence was then amplified from sRAGE transgenic animals with primers to introduce compatible restriction sites and inserted into the scAAV CBh-miR122t-SV4pA vector using MluI (NEB #R3198) and SpeI (NEB #R3133) restriction enzymes. The resulting plasmid was verified with NGS for correct sequence and ITR integrity.

## AAV production

All viral vectors were produced using a triple plasmid transient transfection method outlined previously [30,31]. First, we grew HEK293 cells in 15 cm tissue culture dishes and cultured until 80% confluent in DMEM containing 10% FBS (Gibco #26140079) and 1% PenStrep (Thermo Fisher #15140122). We expanded the cells in 4X15cm dishes followed by seeding them into a HYPERFlask (Millipore Sigma #CLS10031-4EA) and expanded to 80% confluency. Cells were then triple transfected with the scAAV-Cbh-sRAGE expression vector, AAV9 rep/cap (Addgene #112862) and Ad helper plasmid (pAd delta F6 from UPenn) at a ratio of 1:1:2 (130 μg:130 μg:260 μg per HYPERFlask) using PEI Max 40000, pH 7.1 (Polysciences, Inc. #24765-1) at a ratio 1.375: 1 of PEI: total DNA. After four days of incubation, we took the supernatant from a HYPERFlask and transferred it to a 1 liter flask and mixed with 3 ml Triton- X 100 (Millipore Sigma #8787-100ML), 56 μl of 10% Pluronic F-68 (Thermo Fisher #24040032), 2.5 mg RNAse A at 1 mg/ml concentration (Millipore Sigma #10109142001),

and 25 U/ml of Turbonuclease (VitaScientific #ACGC80007). We then took the mixture and poured back into the HYPERFlask and lysed cells and eliminated plasmid DNA by placing the samples on an orbital shaker (150 rpm) at 37 °C for 1 hour. Next, we transferred lysate to a separate container, and the HYPERFlask was washed with 140 ml of DPBS (Life Tech #10010072) and subsequently mixed with the lysate. Lastly, we centrifuged the total lysate mixture at 4000 g, 4 °C for 30 min. The supernatant was then strained through a 0.45 μm PES bottle-top filter (Thermo Fisher #295- 4545) before loading onto a high- performance liquid chromatography (HPLC) machine.

## High performance liquid chromatography

AAVs were purified according to published protocols [30]. In brief, "AAVs were affinity-enriched with either pre-packed AAVX POROS CaptureSelect resin columns (Thermo Fisher #A36652), or with free AAVX resin (Thermo Fisher #A36741), packed into 6.6 mm X 100 mm column (Glass, Omnifit, kinesis-USA). Columns were affixed to an AKTA Pure 25 liter HPLC system (GE Life Sciences #29018224) equipped with an auxiliary sample pump S9 (GE Life Sciences #29027745). All procedures were performed at room temperature (approximately 24 °C). Regardless of the volume of resin used, column volume ([CV]) was limited to 1mL for each purification. Before applying AAV lysate, the column was pre-equilibrated with wash buffer 1X Tris-buffered Saline (1X TBS) (Boston Bioproducts). Lysate was purified at most 1 day before loading onto the HPLC and warmed to room temperature before loading. Lysate was loaded at a flow-rate-to-resin-volume ratio which allowed for ~2 minute residence time on the resin (either 1 ml of resin and a flow rate of 0.5 ml/min, or 4 ml resin with a flow rate of 2 ml/min). When performing purifications with 1 ml of resin, the column with bound AAV was rinsed with 10 [CV] of 1X TBS, followed by washes of 5 [CV] of 2X TBS, 10 [CV] 20% ethanol and 10 [CV] 1X TBS wash. Bound AAV was eluted with a low-pH (pH 2 to 2.9) buffer of 0.2 M Glycine in 1X TBS, containing 0.01% vol/vol Pluronic F-68 at a rate of 1 ml/minute. Resin was then rinsed with 10 [CV] of 1X TBS regenerated with 15 [CV] 0.1M phosphoric acid (pH 1) and 15 [CV] 6 M guanidine HCl at flow rate of 1 ml/min and rinsed again with 10 [CV] 20% ethanol and 10 [CV] 1X TBS. Elution fractions were collected as 1 ml volumes per fraction. To neutralize the eluted vector solution, 1 M Tris-HCL (pH 8.0) was added directly into the fraction collection tube prior to elution at 1/10th of the fraction volume. Peak fractions were collected based on UV (280 nm) absorption graphs, filter sterilized using 0.2 μm PES syringe filters (Corning #431229), and buffer exchanged using Amicon Stirred Cell (Merck Millipore #UFSC05001) concentrators with a molecular weight cut-off of 50 kDa or 100 kDa (EMD Millipore #UFC905008). High-purity nitrogen gas (NI UHP80 Airgas) was employed as a pressure source (40–70 psi). All plasticware and tips were pre-coated with the AAV formulation buffer (FFB: 1xPBS, 35mM NaCl and 0.001% Pluronic) for ~15 minutes at room temperature before applying AAV containing solutions at any step of the purification process".

## AAV titration

AAV titer was determined by qPCR as previously described [32]. PCR reactions were run on a 7500 Real-Time PCR System (Applied Biosystems, Foster City, CA, USA), using NEB Luna Universal Probe qPCR master mix (NEB #M3004S) with bGHpA-targeted primer probes (bGH-Forward: GCCAGCCATCTGTTGT, bGH-Reverse: GGAGTGGCACCTTCCA, bGH-Probe: 6FAM- TCCCCCGTGCCTTCCTTGACC-TAMRA). We employed linearized scAAV-EGFP DNA at a series of 10-fold dilutions of known concentration as a standard. After 95°C holding stage for 10 seconds, two-step PCR cycling stage was performed at 95 °C for 5

seconds, followed by 60 °C for 5 seconds for 40 cycles. Genomic vector titers were interpolated from the standard. Additionally, Cbh-targeted primer-probes (CAG3-F: TCAATGACGGTA-AATGGCCCG, CAG3-Rev CACCTCGACCATGGTAATAGCG, CAG3-Probe 6FAM-CCTGGCATTATGCCCAGTACATGACC-TAMRA) were used as secondary primer probes for titer verification.

### AAV in vivo studies

All animal procedures were performed with the approval of the Institutional Animal Care and Use Committee (IACUC) of the Harvard Office of Animal Resources. Animals in all experiments were injected with AAV9-sRAGE or vehicle (FFB: 1xPBS, 35mM NaCl and 0.001% Pluronic) retro-orbitally at 1e11 vg per mouse. All injections were carried out using a total volume of 150 µl and mice were anesthetized using vaporized isofluorane prior to injection. Immediately prior to injection, the injected eye was cleaned using ophthalmic betadine (HVS 900644_-RX), washed using normal saline 0.9% (Patterson Vet 07-887-0441), and immediately after injection a drop of Proparacaine HCL ophthalmic solution (Patterson Vet 07- 885-9765) was applied to minimize discomfort and scratching of the injection site.

### Inverted grid hanging

To test for muscle strength using the inverted grid method, mice were placed on a wire grid (42 x 29 cm in size, with 1mm diameter wire thickness, and modified with 10 cm horizontal flaps attached to all edges to prevent mice from climbing on top of the grid), and gently inverted approximately 80 cm above a fall bed made of soft material, and a timer was started. Mice were left to hang until they fell onto the bedding, then quickly (within 3 seconds) placed back on the grid and inverted to continue hanging. After the third fall, timer was stopped and the time was recorded. On the day prior to testing, mice were trained on the inverted grid for 2 min, 3 min and 3min with at least 20 min rest in between each set. Then, on the testing day, mice were allowed to hang until three falls as described above. This was repeated twice, with at least 1 hour between testing to allow for sufficient recovery, and the average of the two measurements reported in the figure. The grid was cleaned with ethanol between every test to remove any smell from previously tested mouse. Animals were allowed to acclimatize for at least 30 minutes to the test room, before testing. Testing was carried out within the circadian light phase, between 12–6pm.

### Treadmill endurance test

Mice were introduced to running on a motorized rodent treadmill set (Maze Eng INC, USA) with zero incline. On the first day of introduction, Mice were placed on the treadmill for 10 min with the belt speed set to 10m/min. To motivate the mice to run, transient electric stimulation was introduced through shock grids present at the front of treadmill platform. The following day, mice underwent an exhaustion fatigue test which involved treadmill moving at a speed of 15m/min for 35 min, followed by 25 min at 20m/min. After 60 min, the treadmill speed was increased to 25m/min and allowed the mice to run until they could no longer sustain the exercise/run. Running time (min) and distance covered (m) were noted at the end of the run.

### Statistical analysis and figures

Statistical analyses including two-tailed unpaired t-tests, one-way ANOVA with Tukey post hoc tests, and two-way ANOVA with Tukey post hoc tests were performed, and figures

generated, using GraphPad Prism 9. Illustrations were generated using BioRender. Data are presented as mean ± standard deviation (SD). Statistical significance throughout was set at P-values less than 0.05, using either one-way ANOVA with Tukey post hoc test, two-way ANOVA with Tukey post hoc test, Student's two-tailed unpaired t test, Mann-Whitney U test, or Kruskal-Wallis test, as indicated in the figure legends.

## Results

### sRAGE accelerates regeneration in aged skeletal muscle after cryoinjury.

To better define the expression of sRAGE throughout the lifespan of mice, we used ELISA to quantify the total amounts of sRAGE in mice ranging from 4- to 18-months of age. Overall, we found that sRAGE levels were similar in the bloodstream of adult and aged mice (Fig 1A). These data suggest that genetically encoded sRAGE is not increasing in the blood during chronological aging as a mechanism to sponge out the negative effects of Advanced Glycation End Products (AGEs) and other ligands—which are known to accumulate in aged blood and muscle and may negatively impact muscle health [19,33]. However, we began to speculate whether increasing the levels of sRAGE in the blood and muscle environment of young and old mice beyond what is naturally being produced may benefit muscle healing after myo-trauma. Therefore, we decided to boost the levels of sRAGE in young and aged mice using a newly developed adeno-associated viral vector 9 (AAV9) delivery system carrying a soluble sRAGE transgene (herein referred to as AAV9-sRAGE) (Fig 1B). In this system, infected cells will have the endogenous *Ager* locus (i.e., RAGE) left intact while also having elevated levels of circulating sRAGE, making it possible to sequester over-active RAGE ligands, and thus potentially improving muscle recovery after injury.

To test this hypothesis, we administered AAV9-sRAGE or AAV formulation buffer (FFB) (herein referred to as vehicle control) to both adult (7–8 months-old) and aged (18 month-old) mice at $1e+11$ vg/mouse and assessed the effects on resting and regenerating skeletal muscle 30 days after the initial viral delivery via retro-orbital injections (Fig 1C). To validate our system, we confirmed that in all mice (adult and aged) injected with AAV9-sRAGE we saw an approximate 10-fold increase in the levels of sRAGE in the blood serum as early as 5 days post infection that stayed elevated over the next 30 days relative to vehicle controls (Fig 1D and G). Next, to assess the impact on muscle repair in AAV9-sRAGE and vehicle control treated mice, we performed a cryo-injury, where a frozen metal rod was pressed against the tibialis anterior (TA) muscle and allowed to recover for 1 week. Interestingly, while the mean cross-sectional area (CSA) of centrally nucleated fibers was similar between adult mice treated with AAV9-sRAGE and vehicle control at 7 days post injury (dpi) (Fig 1E and F), in aged mice, we noticed an increase in fiber sizes suggesting enhanced muscle healing when aged mice were treated with AAV9-sRAGE relative to vehicle controls (Fig 1H and I). Lastly, to test whether the observed regenerative benefit of sRAGE extends to other injury models, we repeated our viral infections with AAV9-sRAGE or vehicle control and performed a barium chloride injury followed by assessment of myofiber sizes 7 dpi. Surprisingly, we did not see enhanced myofiber repair after $BaCl_2$ injury in mice treated with AAV9-sRAGE relative to vehicle controla regardless of age, suggesting a context-specific role for sRAGE in the regenerative process (S1 Fig A–F)

We also measured the impact of sRAGE supplementation on muscle function and metabolism of aged mice receiving AAV9-sRAGE or vehicle control. Thirty days after administration of AAV9-sRAGE or vehicle control, we found that treated mice showed no differences in overall body weight (Fig 1J), muscle strength measured by a wire hang test (Fig 1K), or endurance running on a treadmill (Fig 1L). However, mice receiving sRAGE-AAV9 did exhibit a

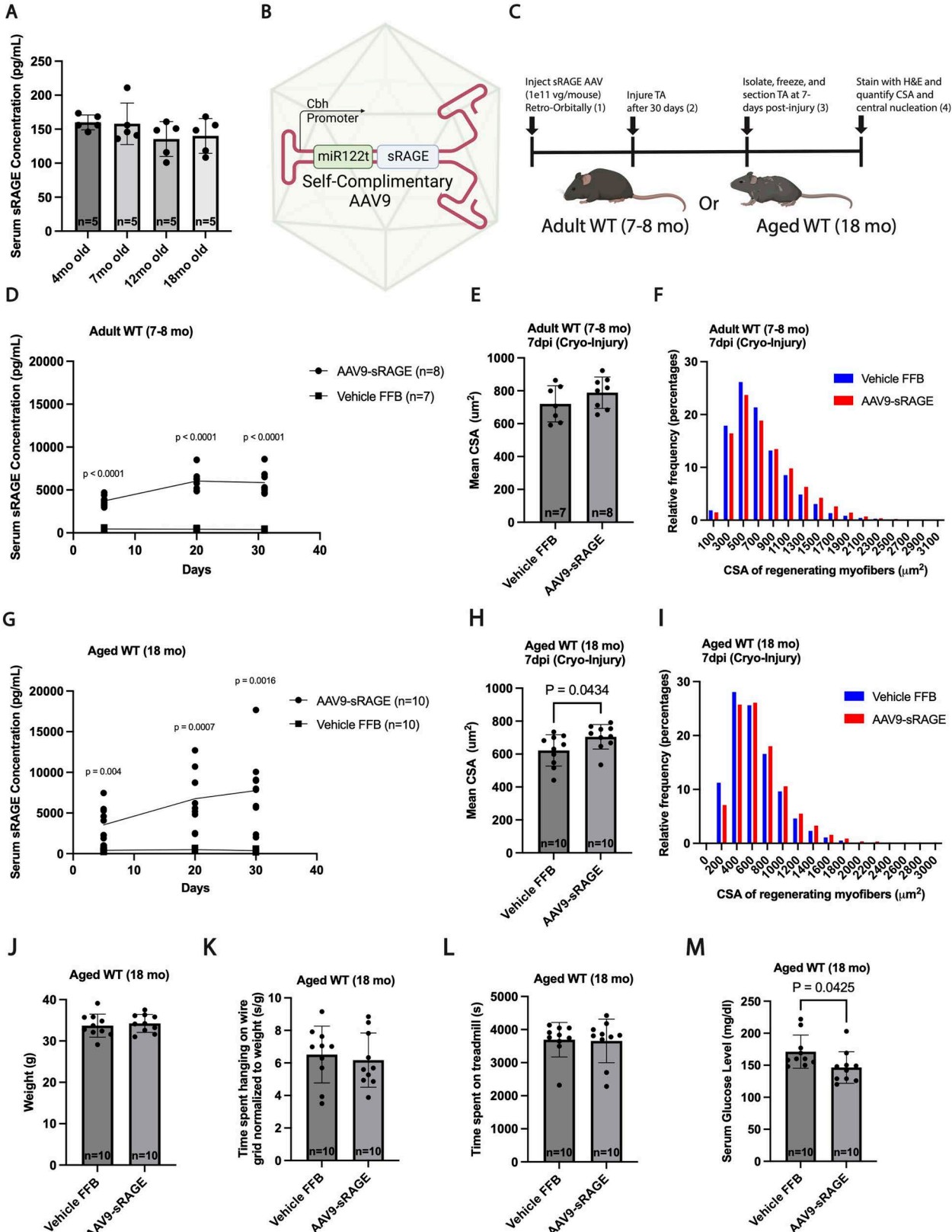

**Fig 1. sRAGE accelerates regeneration in aged skeletal muscle after cryoinjury.** A) ELISA measurement of serum sRAGE concentration in 4, 7, 12 and 18 month old C57BL/6J mice (n = 5 male mice per condition). B) Design of AAV9-sRAGE viral vector delivery system comprising of

a truncated constitutive CBA hybrid (CBh) promoter, an miRNA-122 target site – to decrease liver targeted expression – alongside the sRAGE transgene, all packaged in a self-complementary backbone. C) Experimental scheme for muscle regeneration assay in vehicle (FBB) and AAV9-sRAGE injected Adult (7–8 months old), and aged (18 months old) mice. D) In vivo kinetics of serum sRAGE levels in vehicle (FBB) and AAV9-sRAGE injected adult mice (n ≥ 7 male mice per condition). Enumeration of the mean CSA (E) and distribution (F) of regenerating (centrally nucleated) muscle fibers in vehicle (FBB) and sRAGE treated, adult mice at 7 days after cryoinjury (n ≥ 7 male mice per condition). G) In vivo kinetics of serum sRAGE levels in vehicle (FFB) and AAV9-sRAGE injected aged mice (n = 10 male mice per condition). H & I) Enumeration of the mean CSA (H) and distribution (I) of regenerating (centrally nucleated) muscle fibers in vehicle (FFB) and sRAGE treated, aged mice at 7 days post cryoinjury (n = 10 male mice per condition). Quantification of body weight (J), grid hang time (K), exercise endurance (L), and serum glucose levels (M) in vehicle (FFB) and sRAGE treated, aged (18 months old) mice. (n = 10 male mice per condition). Dots represent data for individual animals overlaid with mean ± SD. Data analyzed for statistical significance by one-way ANOVA with Tukey post hoc test (A, D, G) or Student's two-tailed unpaired t test (E, H, J, K, L, M). Myofiber size distributions analyzed by Mann-Whitney U test (F, I).

significant reduction in serum glucose levels (Fig 1M). Overall, these data suggest that while specific supplementation of sRAGE does not alter muscle endurance and strength of aged mice, it can enhance the regenerative response after cryo-damage and modulate glucose metabolism in aging skeletal muscle.

## sRAGE enhances skeletal muscle regeneration after cryo-damage in a mouse model of atherosclerosis

Given that aged mice and mice with diabetes and atherosclerosis both display impaired muscle regeneration, we next evaluated whether augmenting sRAGE could also enhance regeneration in mice genetically disposed to disease. To test this hypothesis, we first administered sRAGE-AAV9 and vehicle control to 3 month-old leptin receptor–mutant (*db/db*) mice and wild-type controls (Fig 2A). Db/db mice are obese and one of the most widely used mouse models of type 2 diabetes. After delivery of sRAGE-AAV9 virus and vehicle control to both groups of mice, we first confirmed that *db/db* mice receiving sRAGE-AAV9 (relative to vehicle control) showed an appreciable increase in the levels of sRAGE starting at day 5 and persisting until day 30 post initial viral delivery (Fig 2B). We found that *db/db* mice treated with sRAGE-AAV9 had similar weights and blood glucose as *db/db* mice treated with vehicle controls, indicating that elevating sRAGE does not reverse the increase in blood glucose and weight observed in *db/db* when compared to healthy controls (Fig 2C and D). As expected, *db/db* mice display impaired regeneration 7dpi relative to healthy WT mice (Fig 2E and F). However, while there was a trend towards larger fiber sizes in *db/db* mice treated with sRAGE-AAV9 (relative to vehicle controls), the increase never reached statistical significance, suggesting that sRAGE overexpression via gene therapy may not be sufficient to rescue the impairments in muscle regeneration associated with diabetic conditions (Fig 2E and F). Similarly, while we could recapitulate the expected impairments in regeneration displayed in *db/db* mice relative to WT mice via a BaCl$_2$-mediated injury, we saw no difference in myofiber sizes 7 dpi between sRAGE-AAV9 virus and vehicle control treated mice injured via BaCl$_2$ (S2 Fig A–C).

We also administered sRAGE-AAV9 to 2–3 month-old mice with a genetic deletion of the apolipoprotein E gene (ApoE$^{-/-}$) (Fig 2G), which results in profoundly hypercholesterolemic animals that rapidly develop atherosclerotic plaques [2,34]. As in previous experiments, we verified expression of the sRAGE transgene in serum starting at 5 days-post-injection and showed further elevated levels at 20 and 30 days-post-injection for the AAV9-sRAGE injected group, relative to vehicle controls (Fig 2H). ApoE$^{-/-}$ mice exhibited no significant differences in weight or blood glucose between AAV9-sRAGE and vehicle treated controls (Fig 2I and J). However, interestingly, analysis of cryo-injured skeletal muscle 7 dpi revealed a mean cross-sectional area that was larger in ApoE$^{-/-}$ mice treated with AAV9-sRAGE when compared to vehicle treated controls (Fig 2K and L). Finally, similar to what we found in aged

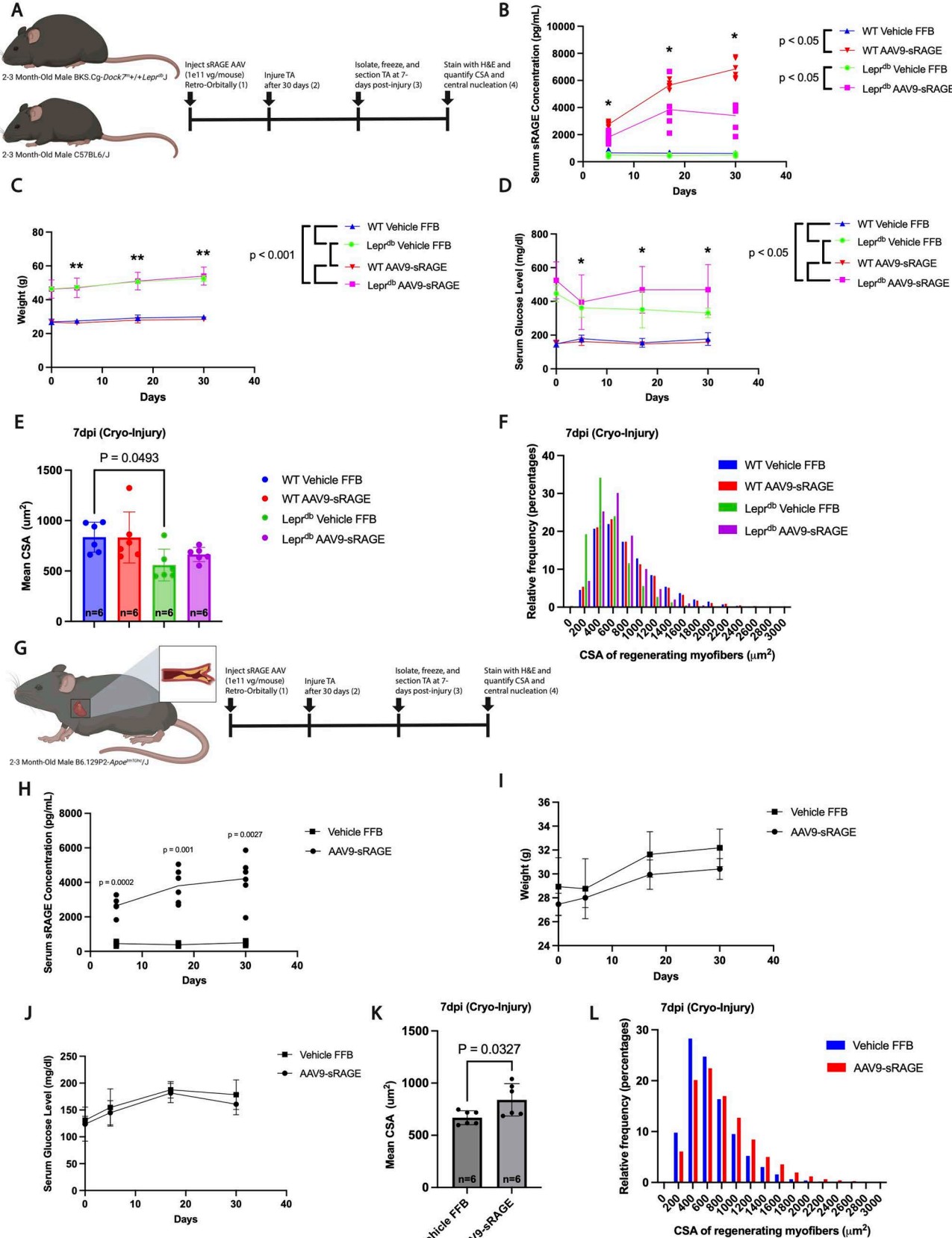

**Fig 2. sRAGE enhances skeletal muscle regeneration after cryo-damage in a mouse model of atherosclerosis.** Experimental scheme for muscle regeneration assay in vehicle (FBB) and AAV9-sRAGE injected diabetic (Lepr^db) and wild-type (WT) mice. B) In vivo kinetics of serum sRAGE levels in

AAV9-sRAGE or vehicle (FFB) injected diabetic (Lepr^db) and wild-type (WT) mice (n = 6 male mice per condition). C) Body weight of diabetic (Lepr^db) and wild-type (WT) animals following vehicle (FFB) or AAV9-sRAGE administration (n = 6 male mice per condition). D) Serum glucose levels of diabetic (Lepr^db) and wild-type (WT) animals following vehicle (FFB) or AAV9-sRAGE administration (n = 6 male mice per condition). Black brackets are shown for those comparisons that are statistically significant. Enumeration of the mean CSA (E) and distribution (F) of regenerating (centrally nucleated) muscle fibers in vehicle (FFB) and AAV9-sRAGE treated, diabetic (Lepr^db) and wild-type (WT) mice at 7 days after cryoinjury (n = 6 male mice per condition). G) Experimental scheme for muscle regeneration assay in vehicle (FBB) and AAV9-sRAGE injected atherosclerotic (ApoE-null) and wild-type (WT) mice. H) In vivo kinetics of serum sRAGE levels in vehicle (FFB) and AAV9-sRAGE injected atherosclerotic (ApoE-null) animals (n = 6 male mice per condition). Body weight (I) and serum glucose levels (J) of atherosclerotic (ApoE-null) mice following vehicle (FFB) and AAV9-sRAGE administration (n = 6 male mice). Enumeration of the mean CSA (K) and distribution (L) of regenerating (centrally nucleated) muscle fibers in vehicle (FFB) and AAV9-sRAGE treated, atherosclerotic (ApoE-null) mice at 7 days after cryoinjury (n = 6 male mice per condition). Dots represent data for individual animals overlaid with mean ± SD. Data analyzed for statistical significance by two-way ANOVA with Tukey post hoc test (B, C, D), one way ANOVA with Tukey post hoc test (E, H, I, J) or Student's two-tailed unpaired t test (K). Myofiber size distributions analyzed by Mann-Whitney U test and Kruskal-Wallis test (F, L). All mice were 2–3 months of age at the time of study entry.

mice (S1 Fig D–F), we observed no change in myofiber sizes 7 days after BaCl₂ in ApoE-null mice treated with AAV9-sRAGE relative to vehicle control ( S2 Fig D-F). Altogether, these results suggest that sRAGE supplementation has no impact on muscle repair in diabetic mice but can enhance muscle healing specifically after cryo-injury in mouse models of high cholesterol leading to atherosclerosis.

## Endogenous and lifelong expression of a transgene encoding sRAGE enhances skeletal muscle regeneration post cryoinjury

To explore how genetically altering the levels of RAGE and sRAGE at the endogenous Ager locus impact muscle regeneration, we next utilized complementary models in which the native full-length RAGE locus is either completely disrupted (Ager^-) or replaced with an sRAGE transgene (Ager^s) (Fig 3A). As expected[27], both Ager^s/+ and Ager^s/s mice showed significantly elevated levels of sRAGE in serum (Fig 3B). Likewise, only mice containing one or more native AGER alleles exhibited expression of c-terminal residues of RAGE via western blot (Fig 3C).

Prior to injury, no differences in skeletal muscle fiber sizes were observed when comparing Ager^s/+, Ager^s/s and wild type animals (Fig 3D). However, examination of regenerating muscle 7 days after cryoinjury (Fig 3E), revealed larger myofiber sizes in both AGER^s/s and AGER^s/+ animals as compared to wild-type mice (Fig 3F). Specifically, at 7 dpi, the mean CSA of regenerating (centrally-nucleated) myofibers in adult AGER^s/s and AGER^s/+ animals was ~30% greater than regenerating fiber CSA in equivalently injured wild-type mice (Fig 3G and H). However, similar to results in the previously discussed models (S1–S2 Fig ), AGER^s/s and AGER^s/+ animals injured via intramuscular injection of BaCl₂ did not show enhanced regenerative activity when compared to analogously injured wild-type mice (S3 Fig A–C).

Lastly, given the central role of MuSCs in the repair of injured muscle, we sought to distinguish cell-intrinsic from cell-extrinsic contributions of sRAGE production in this cell type via a transplantation model [35]. Thus, we transplanted 3000 or 6000 MuSCs harvested from each of the six genotypes of Ager transgenic/knockout animals into the pre-injured (BaCl₂) muscles of dystrophic mdx recipient mice. The total number of donor MuSC-derived dystrophin + fibers did not differ across the various transgenic donor lines when assessed at three weeks post-transplantation (Fig 3I and J). Altogether, these data suggest that sRAGE production in MuSCs alone is not sufficient to enhance skeletal muscle repair, but rather, sRAGE production in the micro-environment is likely key to its benefit to muscle healing after injury.

## Discussion

Our study is the first to highlight a functional role for sRAGE as an accelerator of muscle repair after cryo-damage. Specifically, we observed significant increases in the mean fiber

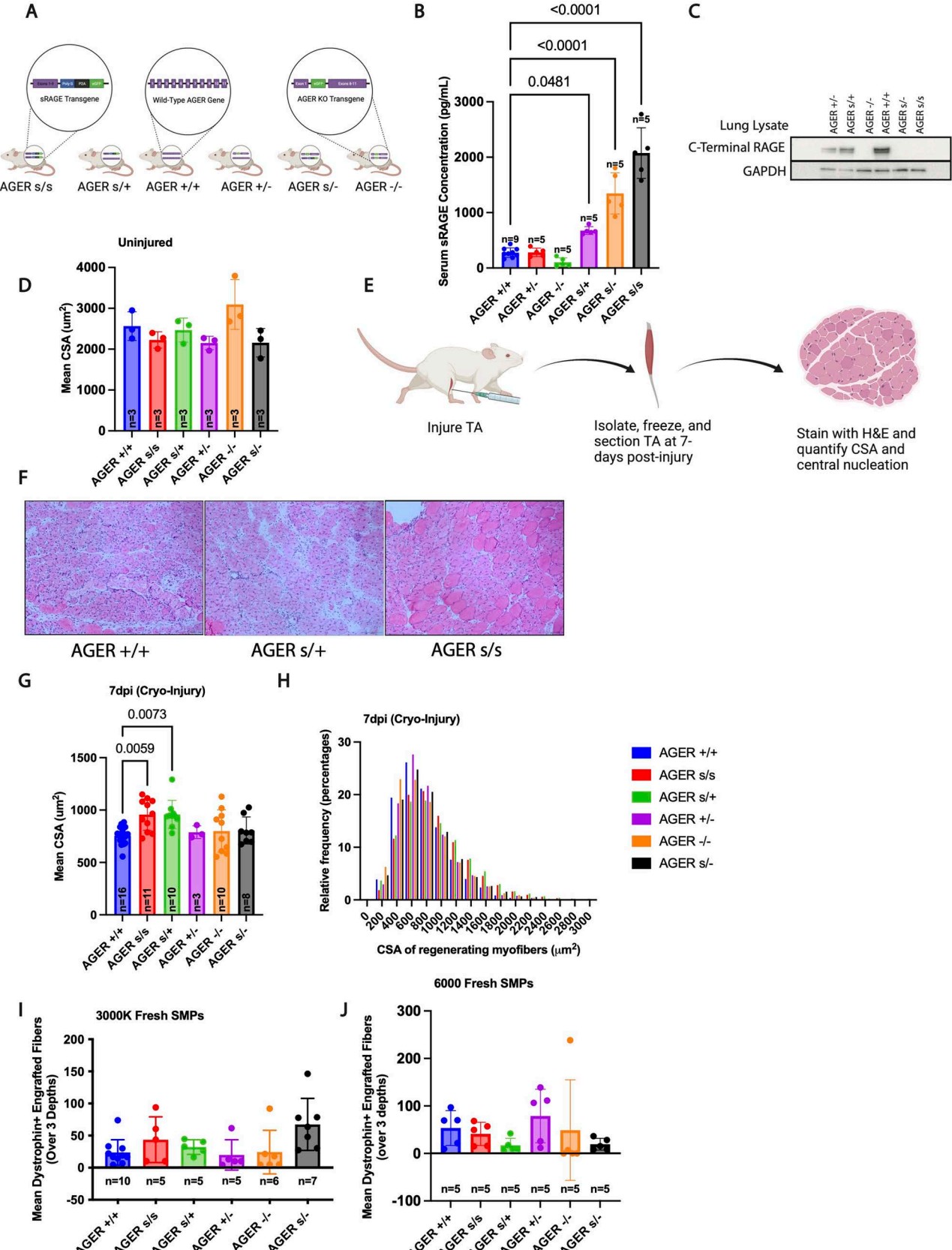

**Fig 3. Endogenous and lifelong expression of a transgene encoding sRAGE enhances skeletal muscle regeneration post cryoinjury.** A) Genetic composition at the *AGER* locus of transgenic and wild type mice used in this study. AGER KO animals were previously generated via a cre-mediated

loxP recombination of essential elements (exons 2 to 7) of the murine RAGE gene. sRAGE transgenic animals were previously generated by replacing exons 10 – 11 of the Ager gene, which encode the transmembrane and cytosolic signaling domains of the murine RAGE, with a bicistronic control element 2A-linked EGFP expression cassette. Therefore, this animal is a full-length RAGE-null and the resulting sRAGE transgene is expressed in RAGE-expressing cell types using its normal gene regulatory elements. B) ELISA measurement of serum sRAGE concentration in mice of the indicated genotypes (n ≥ 5 male mice per condition). **C)** Confirmation of C-terminal RAGE expression in mice bearing at least one *Ager+* allele, and absence in mice bearing only *Ager*s or *Ager-* alleles, via Western blot. D) Enumeration of the mean CSA of muscle fibers in uninjured mice of the indicated genotypes (n = 3 male mice per condition). E) Experimental scheme for muscle regeneration assay in RAGE knockout and transgenic mice. F) Representative H&E stained images of regenerating muscle from *Ager+/+*, *Ager*s/+ and *Ager*s/s mice subjected 7 days previously to cryoinjury. Enumeration of the mean CSA (G) and distribution (H) of regenerating (centrally nucleated) muscle fibers in wild-type, knockout and transgenic mice at 7 days after cryoinjury (n ≥ 3 male mice per condition). Quantification of engrafted dystrophin+ muscle fibers following transplantation of 3000 (I) or 6000 **(J)** Sca1-; Cd45-; Cd11b-, Ter119-, Cd31-; ItgB7+; CXCR4+ MuSCs from the indicated genotypes of mice (wild-type, knockout or transgenic) into the pre-injured muscles of *mdx* recipient mice (n ≥ 5 male mice per condition). Dots represent data for individual animals overlaid with mean ± SD. Data analyzed for statistical significance by one-way ANOVA with Tukey post hoc test (B, D, G, I, J). Myofiber size distributions analyzed by Kruskal-Wallis test (H). All mice were 7–8 months of age at the time of study entry.

sizes 7 days after cryo-injury in adult, aged, and diseased mice when sRAGE was increased in vivo using two complementary animal model systems: (1) AAV delivery of sRAGE (Fig 1 and 2) and (2) life-long forced sRAGE expression in a transgenic mouse line (Fig 3). That both genetic systems gave similar results highlight the universal importance of sRAGE in muscle healing after a freeze injury.

What is the mechanism for how boosting sRAGE enhances muscle repair after cryo-damage? Earlier work has shown that RAGE ligands are chronically activated in aged MuSCs—which display reduced proliferation and differentiation [36–38]. In fact, knockdown of s100B in aged MuSCs even reverses some of the aged phenotype [38]. Although more work in this area is needed, we suspect that in our system, sRAGE likely acts as a soluble decoy to sequester RAGE ligands, and thus preventing their negative impact on muscle repair in the elderly. While recent work suggests that RAGE/sRAGE directly interacts with various ligands known to impact muscle regeneration such as s100b, HMGB1, C1q [10,12,36,39], the role of AGEs known to interact with RAGE such as Carboxymethyl-lysine, Carboxyethyl-lysine, Methylglyoxal, Glyoxal, Pyrraline, and Pentosidine [40–45] are not as well studied in the context of skeletal muscle regeneration. Advances in technology will be needed to accurately measure AGE variants and determine (1) their abundance across lifespan and (2) determine which are most relevant for skeletal muscle pathology. Additionally, future work to profile the multitude of other RAGE ligands in young and old skeletal muscle as well as identifying the key cellular source that releases these products into the micro-environment is needed to fully uncover the mechanism for how sRAGE boosts muscle repair in aging animals.

Previous studies have revealed that RAGE and its ligands influence insulin resistance and skeletal muscle atrophy in diabetic contexts [13,46]. In this scenario, sRAGE may also dampen intracellular signaling through RAGE and thereby ameliorate diabetic insulin resistance and muscle atrophy [47]. Although not statistically significant, we found that regenerating skeletal muscle fibers from diabetic animals trended towards having larger muscle fibers after cryo-injury when treated with AAV9-sRAGE relative to vehicle controls (Fig 2E and F). Similarly, previous findings showed that, in atherosclerosis-prone mice devoid of ApoE, treatment with antagonists of RAGE or its genetic deletion reduced atherosclerotic lesion area and macro-phage content [38]. These data are consistent with our findings that supplementing mice with sRAGE via AAV delivery significantly improves muscle regeneration after cryoinjury in ApoE-/- mice, possibly by working through a similar mechanism of 'sponging' out RAGE ligands that would otherwise cause atherosclerotic lesions and reduced vascular modeling after injury.

It is worth noting that previous work revealed that full length RAGE expression is required for proper stem cell activation after injury, at least in younger animals (2–3 months old) [10]. In our study, our *AGER*s/s mice (which also lack endogenous RAGE) display enhanced

regeneration after cryo-damage, suggesting the possibility that sRAGE could functionally compensate for full length RAGE both in the developmental trajectory of skeletal muscle as well as after acute injury. However, we note that we were not able to reproduce the regenerative delays characteristic of 2–3 month old RAGE KO animals [10] in our older adult cohort (7–8 months old), indicating that RAGE expression may be required for proper regenerative response only at earlier life stages.

Given their central role in the repair of injured muscle, we also evaluated the cell-intrinsic role of sRAGE expression on MuSCs through cell transplantations, which test the ability of these cells to engraft and contribute to the repair of the recipient muscle. Our results showed that regardless of cell number, MuSCs expressing sRAGE engrafted muscle in a similar way as wild-type MuSCs. These data suggest that sRAGE impact on MuSC activity may be through modulation of the extracellular environment that works to impede muscle repair in aging skeletal muscle.

Our results clearly demonstrate that sRAGEs ability to boost a regenerative response depends on the injury model employed. How is this possible? Well, it has been reported that freeze and $BaCl_2$ injuries progress quite differently during the regenerative process and we highlight several of these differences below [48]. First, $BaCl_2$ injury follows a sequential and synchronous regeneration pattern characterized by neutrophil and macrophage infiltration, whereas a freeze injury is characterized by asynchronous regeneration where different regeneration steps are present simultaneously [48]. Thus, differences in the composition of infiltrating immune cells—which can produce RAGE ligands [49,50]—may be different after $BaCl_2$ versus freeze injury. Second, $BaCl_2$ injury displays a more moderate disruption in vasculature that requires less time to fully restore, whereas freeze injury leads to near total destruction of the vascular network—which may lead to more blood-borne RAGE ligands (including cytokines) getting secreted into the muscle environment. Therefore, future work should focus on further characterizing how sRAGE/RAGE and its ligands change in young and old mice after $BaCl_2$ and Freeze injury, and how administering sRAGE impacts downstream RAGE signaling and muscle regenerative function.

Finally, we highlight that most papers in the field of muscle biology only perform one type of injury model in each study. This reality begs the question of whether some previously published phenotypes may be variable if put to test with multiple injury models. To date, there are at least 6 major injury models used in most studies including freeze, $BaCl_2$, cardiotoxin, notexin, crush injury, or multiple needle stick injuries, with labs usually choosing a model based on convenience or precedence (i.e., which lab they were trained in). We suspect that as we learn more about the mechanism of each injury model, each investigator will choose the injury model that best suits the biology and question they are trying to solve. With this new mindset, it should not be surprising or worrisome when a phenotype is seen with one model and not the other, especially since recent evidence suggests that each of these injury models initiates a distinct stem cell activation and repair response. Thus, we hope that our study will caution all investigators in the muscle field to choose their injury model carefully

## Conclusion

Our study demonstrates that, in addition to the well documented role that sRAGE plays in ameliorating pathologies across various non-muscle tissues, sRAGE also can serve as an accelerator of regeneration following freeze insults to the skeletal muscle. These results raise the interesting possibility that administration of sRAGE in some situations may be a potential therapeutic strategy for the treatment of skeletal muscle injury, aging muscle, and degenerative disorders. We hope that our data reported here from multiple injury models and genetic systems for modulating endogenous RAGE signaling will prove useful for guiding

appropriate deployment of RAGE/sRAGE targeting therapeutics in ongoing and future clinical investigations.

## Supporting information

**S1 Fig. sRAGE does not alter regeneration in adult and aged skeletal muscle after Barium Chloride injury.** A) Experimental scheme for muscle regeneration assay in vehicle (FBB) and AAV9-sRAGE injected adult (7-8 months old) mice. B & C) Enumeration of the mean CSA (B) and distribution (C) of regenerating (centrally nucleated) muscle fibers in vehicle (FFB) and sRAGE treated, adult (7-8 months old) mice at 7 days after $BaCl_2$ induced injury ($n \geq 7$ male mice per condition). D) Experimental scheme for muscle regeneration assay in vehicle (FBB) and AAV9-sRAGE injected aged (18 months old) mice. E & F) Enumeration of the mean CSA (E) and distribution (F) of regenerating (centrally nucleated) muscle fibers in vehicle (FFB) and sRAGE treated, aged (18 months old) mice at 7 days after $Bacl_2$ induced injury ($n = 10$ male mice per condition). Dots represent data for individual animals overlaid with mean ± SD. Data analyzed for statistical significance by Student's two-tailed unpaired t test (B, E). Myofiber size distributions analyzed by Mann-Whitney U test (C, F).
(TIF)

**S2 Fig. sRAGE does not enhance skeletal muscle regeneration after BaCl2-damage in mouse models of diabetes and atherosclerosis.** A) Experimental scheme for muscle regeneration assay in vehicle (FBB) and AAV9-sRAGE injected diabetic (Lepr$^{db}$) and wild-type (WT) mice. B & C) Enumeration of the mean CSA (B) and distribution (C) of regenerating (centrally nucleated) muscle fibers in vehicle (FFB) and AAV9-sRAGE treated, diabetic (Lepr$^{db}$) and wild-type (WT) mice at 7 days after $BaCl_2$-induced injury (N = 6 male mice per condition). D) Experimental scheme for muscle regeneration assay in vehicle (FBB) and AAV9-sRAGE injected atherosclerotic (ApoE-null) and wild-type (WT) mice. E & F) Enumeration of the mean CSA (E) and distribution (F) of regenerating (centrally nucleated) muscle fibers in vehicle (FFB) and AAV9-sRAGE treated, atherosclerotic (ApoE-null) mice at 7 days after $BaCl_2$-induced injury (n = 6 mice per condition). Dots represent data for individual animals overlaid with mean ± SD. Data analyzed for statistical significance by one-way ANOVA with Tukey post hoc test (B) and Student's two-tailed unpaired t test (E). Myofiber size distributions analyzed by Mann-Whitney U test and Kruskal-Wallis test (C, F). All mice were 2-3 months of age at the time of study entry.
(TIF)

**S3 Fig. Endogenous and lifelong expression of a transgene encoding sRAGE does not enhance skeletal muscle regeneration post BaCl2-mediated injury.** A) Experimental scheme for muscle regeneration assay in RAGE knockout and transgenic mice. B & C) Enumeration of the mean CSA (B) and distribution (C) of regenerating (centrally nucleated) muscle fibers in wild-type, knockout and transgenic mice at 7 days after $Bacl_2$-induced injury ($n \geq 3$ male mice per condition). Dots represent data for individual animals overlaid with mean ± SD. Data analyzed for statistical significance by one-way ANOVA with Tukey post hoc test (B). Myofiber size distributions analyzed by Kruskal-Wallis test (C). All mice were 7-8 months of age at the time of study entry.
(TIF)

**S1 Raw images. Confirmation of C-terminal RAGE expression in mice bearing at least one *Ager*⁺ allele, and absence in mice bearing only *Agers or Ager⁻* alleles, via Western blot. raw jpg files show western blot images of membranes stained for RAGE protein (1:200, rabbit anti-RAGE, ab3611) and the corresponding GAPDH protein control.**
(PDF)

## Acknowledgements

We thank Dr. Li Lin and Dr. Kathy Perdue for providing sRAGE transgenic animals as well as Dr. Chun Lee for providing RAGE KO animals. We thank J. LaVecchio, S. Ionescu, and N.Kheradmand for assistance with FACS. Our graphics were created using BioRender.

## Author contributions

**Conceptualization:** Naftali Horwitz, Albert E. Almada, Amy J. Wagers.

**Data curation:** Naftali Horwitz, Michael Florea, Medha KC, Tina Liu, Vivian Garcia, Rebekah Kim, Amy Lam, Kathleen Messemer, Christopher Rios, Albert E. Almada, Amy J. Wagers.

**Formal analysis:** Naftali Horwitz, Medha KC, Tina Liu, Albert E. Almada, Amy J. Wagers.

**Funding acquisition:** Naftali Horwitz, Albert E. Almada, Amy J. Wagers.

**Investigation:** Naftali Horwitz, Michael Florea, Medha KC, Tina Liu, Vivian Garcia, Rebekah Kim, Amy Lam, Kathleen Messemer, Christopher Rios, Albert E. Almada, Amy J. Wagers.

**Methodology:** Naftali Horwitz, Michael Florea, Medha KC, Tina Liu, Vivian Garcia, Rebekah Kim, Amy Lam, Kathleen Messemer, Christopher Rios, Albert E. Almada, Amy J. Wagers.

**Project administration:** Naftali Horwitz, Amy J. Wagers.

**Resources:** Albert E. Almada, Amy J. Wagers.

**Software:** Amy J. Wagers.

**Supervision:** Naftali Horwitz, Albert E. Almada, Amy J. Wagers.

**Validation:** Naftali Horwitz, Michael Florea, Amy J. Wagers.

**Visualization:** Naftali Horwitz, Amy J. Wagers.

**Writing – original draft:** Naftali Horwitz, Albert E. Almada, Amy J. Wagers.

**Writing – review & editing:** Naftali Horwitz, Albert E. Almada, Amy J. Wagers.

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
