## [Decision Letter · Decision Letter 0]

4 Dec 2024

PONE-D-24-50348Soluble RAGE enhances muscle regeneration after cryoinjury in aged and diseased micePLOS ONE

Dear Dr. Horwitz,

Thank you for submitting your manuscript to PLOS ONE. After careful consideration, we feel that it has merit but does not fully meet PLOS ONE’s publication criteria as it currently stands. Therefore, we invite you to submit a revised version of the manuscript that addresses the points raised during the review process.

**ACADEMIC EDITOR:**Reviewers have raised concerns about sRAGE levels, the age of the animals, and the areas of muscle examined. Please address all these issues carefully before resubmitting.

We look forward to receiving your revised manuscript.

Kind regards,

Ram Nagaraj

Academic Editor

PLOS ONE

“This study was supported by an NIH training grant award (2T32DK007260) to NH, awards from the Paul F. Glenn Medical Foundation and NIH (R01AG048917) to AJW, and awards from the Burroughs Wellcome Fund, NIH diversity supplement (3R01AG048917-02S1), and NIH (1R01AR080753) to AEA.”

“A.J.W. is a scientific advisor for Kate Therapeutics and Frequency Therapeutics, as well as a co-founder and scientific advisory board member and holds private equity in Elevian, Inc., a company that aims to develop medicines to restore regenerative capacity. Elevian also has provided sponsored research to the Wagers lab.”

Reviewers' comments:

Reviewer's Responses to Questions

**Comments to the Author**

1. Is the manuscript technically sound, and do the data support the conclusions?

Reviewer #1: Yes

Reviewer #2: Yes

2. Has the statistical analysis been performed appropriately and rigorously? 

Reviewer #1: I Don't Know

Reviewer #2: Yes

3. Have the authors made all data underlying the findings in their manuscript fully available?

Reviewer #1: Yes

Reviewer #2: Yes

4. Is the manuscript presented in an intelligible fashion and written in standard English?

Reviewer #1: Yes

Reviewer #2: Yes

5. Review Comments to the Author

Reviewer #1: The authors investigate the impact of soluble receptor for advanced glycation end-products (sRAGE) on the regeneration of skeletal muscle damaged by cryoinjury. They demonstrate that the expression of sRAGE improves muscle regeneration in aged mice and in atherosclerotic mouse models. Although this article contains some interesting aspects in the field of glycation, there are several issues that need to be clarified.

Critique

1. In Figure 1D and G, was there a difference in the serum sRAGE concentration between the Vehicle FFB groups of 7-8-month-old and 18-month-old mice without sRAGE? Does the expression levels of sRAGE vary with age?

2. In Figure 3, why did the authors use 7-8-month-old mice, which did not show differences in CSA (cross-sectional area) as shown in Figure 1?

3. Regarding the labeling of the transgenic mice used in Figure 3A, how about providing an additional explanation in the text or the figure legend?

4. In the Discussion, the authors mention that "Earlier work has shown that RAGE ligands are chronically activated in aged MuSCs—which display reduced proliferation and differentiation [36-38]." Does this imply that sRAGE is released from MuSCs in aged individuals, and a reduction of sRAGE within MuSCs leads to insufficient sequestration of RAGE ligands, thereby negatively affecting muscle repair? Is this consistent with the statement in the Introduction that sRAGE is present in the blood of centenarians?

Minor comments

1. In discussion: “In our study, our AGERs/s mice (which also lack endogenous RAGE) displayS enhanced …”. Is displayS a typo for displays?

2. sRAGE is considered to act as a decoy receptor, preventing ligands bind to RAGE and thereby inhibiting the negative effects mediated by RAGE. Would it be possible to discuss the name of AGEs that bind to RAGE or sRAGE in the main text?

Reviewer #2: Horwitz et al have aimed to study the role of sRAGE in muscle regeneration after cryo/barium chloride injury in healthy, aged and diseased mice models. They have used different mouse strains to design and execute different experiments to address their objective. Overall, this is a well-designed study which provides novel insights. However, there are some concerns in the paper which needs to be addressed

1. In figure 1 (H & I), the authors state that the mean cross-sectional area (CSA) of centrally nucleated fibers increase in aged mice treated with AAV9-sRAGE relative to vehicle. Can it be corroborated whether this is a significant increase? If so, can it be related to sRAGE alone?

2. In figure 3 (I & J), the data suggest that the number of donor MuSC-derived dystropin+ fibers does not differ across different donor lines. The authors do not elaborate about this in the discussion session.

Other clarifications in Materials and methods

3. Western blotting – Dilutions of Both primary and secondary antibody may be mentioned. Also mention the secondary antibody used.

4. A short description of the ELISA protocol may be provided.

5. The make of the restriction enzymes used may be provided

6. Catalog numbers and brands of all chemicals/kits/buffers may be provided

6. PLOS authors have the option to publish the peer review history of their article (what does this mean? ). If published, this will include your full peer review and any attached files.

**Do you want your identity to be public for this peer review?** For information about this choice, including consent withdrawal, please see our Privacy Policy .

Reviewer #1: No

Reviewer #2: No

---

## [Author Response · Author response to Decision Letter 1]

22 Dec 2024

Reviewer #1:

1. In Figure 1D and G, was there a difference in the serum sRAGE concentration between the Vehicle FFB groups of 7-8-month-old and 18-month-old mice without sRAGE? Does the expression levels of sRAGE vary with age?

We thank the reviewer for the opportunity to further explain this point. Similar to the results in Figure1A showing no difference in endogenous sRAGE between 4- to 18-month-old mice (i.e., adult and aged animals), when we plot the FFB injected animals from young and old mice shown in Figure 1D and G on the same plot, we also observe no significant difference in serum sRAGE using a two-way ANOVA with a Šídák's multiple comparisons test. Therefore, we conclude that in sedentary C57BL/6J mice, sRAGE levels in the blood do not naturally change (up or down) during chronological aging.

Please refer to the "Response to Reviewer" file for the relevant plot.

2. In Figure 3, why did the authors use 7-8-month-old mice, which did not show differences in CSA (cross-sectional area) as shown in Figure 1?

We thank the reviewer for this comment. Although we didn’t find significant differences in CSA of skeletal muscle after Freeze injury between adult animals treated with AAV-sRAGE and vehicle control (Figure 1E-F), we did appreciate a trend which we anticipate may reach statistical significance in a model where there is lifelong, endogenous overexpression as seen in the sRAGE transgenic animals (Figure 3). For practical reasons, given the diverse array of mouse lines and breeding schemes required to generate the 6 different lines in Figure 3 (i.e. sRAGE transgenic x wild-type, RAGE KO x wild-type, etc.), we were limited in the scope of our colony, especially once the pandemic forced us to pause our breeding. Although we agree that investigating the older sRAGE transgenic animals would bolster our findings, we nonetheless found a significant increase in muscle CSA following Freeze injury in the adult sRAGE transgenic mice. These data ultimately support our overall hypothesis: that lifelong, endogenous overexpression of sRAGE is beneficial to skeletal muscle regeneration.

3. Regarding the labeling of the transgenic mice used in Figure 3A, how about providing an additional explanation in the text or the figure legend?

We added the following sentence into the text on Pg 12, figure 3 legend, shown below:

“AGER KO animals were previously generated via a cre-mediated loxP recombination of essential elements (exons 2 to 7) of the murine RAGE gene. sRAGE transgenic animals were previously generated by replacing exons 10−11 of the Ager gene, which encode the transmembrane and cytosolic signaling domains of the murine RAGE, with a bicistronic control element 2A-linked EGFP expression cassette. Therefore, this animal is a full-length RAGE-null and the resulting sRAGE transgene is expressed in RAGE-expressing cell types using its normal gene regulatory elements.”

4. In the Discussion, the authors mention that "Earlier work has shown that RAGE ligands are chronically activated in aged MuSCs—which display reduced proliferation and differentiation [36-38]." Does this imply that sRAGE is released from MuSCs in aged individuals, and a reduction of sRAGE within MuSCs leads to insufficient sequestration of RAGE ligands, thereby negatively affecting muscle repair? Is this consistent with the statement in the Introduction that sRAGE is present in the blood of centenarians?

We thank the reviewer for the opportunity to clarify this point further. We believe that the elevation in blood serum concentrations of sRAGE (as seen in centenarians) is likely a compensatory mechanism by the body to sequester RAGE ligands that are chronically activated in aged skeletal muscle (including MuSCs). Whether the “higher amount of sRAGE” that centenarians have is enough to sequester all the harmful RAGE ligands and promote a regenerative benefit is unknown. Regardless, our mouse studies clearly demonstrate that the elderly bodies’ natural amount of sRAGE may not be sufficient, and indeed, boosting sRAGE in aged mammalian skeletal muscle gives the benefit of enhancing the muscle healing process.

5. In discussion: “In our study, our AGERs/s mice (which also lack endogenous RAGE) displayS enhanced …”. Is displayS a typo for displays?

We thank the reviewer for catching this typo. We made the correction in the revised version of manuscript.

6. sRAGE is considered to act as a decoy receptor, preventing ligands bind to RAGE and thereby inhibiting the negative effects mediated by RAGE. Would it be possible to discuss the name of AGEs that bind to RAGE or sRAGE in the main text?

We thank the reviewer for making this suggestion. We added a sentence to the discussion regarding AGE targets of RAGE/sRAGE in the revised version of the manuscript on pg 13.

“While recent work suggests that RAGE/sRAGE directly interacts with various ligands known to impact muscle regeneration such as s100b, HMGB1, C1q [10,12,36,39], the role of AGEs known to interact with RAGE such as Carboxymethyl-lysine, Carboxyethyl-lysine, Methylglyoxal, Glyoxal, Pyrraline, and Pentosidine [40-45] are not as well studied in the context of skeletal muscle regeneration. Advances in technology will be needed to accurately measure AGE variants and determine (1) their abundance across lifespan and (2) determine which are most relevant for skeletal muscle pathology.”

Reviewer #2

1. In figure 1 (H & I), the authors state that the mean cross-sectional area (CSA) of centrally nucleated fibers increase in aged mice treated with AAV9-sRAGE relative to vehicle. Can it be corroborated whether this is a significant increase? If so, can it be related to sRAGE alone?

We welcome the opportunity to further clarify this point. Our studies found that the (1) mean cross sectional area (CSA) of myofibers (Two-Tailed, unpaired, T-test) and (2) the distribution of all centrally nucleated myofibers (CNFs) (using Mann-Whitney) are larger in the AAV9-sRAGE aged mice when compared to FFB aged control mice. These data are consistent with the idea that sRAGE supplementation enhanced muscle healing after a freeze injury.

2. In figure 3 (I & J), the data suggest that the number of donor MuSC-derived dystropin+ fibers does not differ across different donor lines. The authors do not elaborate about this in the discussion session.

We thank the reviewer for pointing this out. We have included more discussion in the discussion section of our revised manuscript on pg 13. The included text is reproduced below:

“Given their central role in the repair of injured muscle, we also evaluated the cell-intrinsic role of sRAGE expression on MuSCs through cell transplantations, which test the ability of these cells to engraft and contribute to the repair of the recipient muscle. Our results showed that regardless of cell number, MuSCs expressing sRAGE engrafted muscle in a similar way as wild-type MuSCs. These data suggest that sRAGE impact on MuSC activity may be through modulation of the extracellular environment that works to impede muscle repair in aging skeletal muscle.”

3. Western blotting – Dilutions of Both primary and secondary antibody may be mentioned. Also mention the secondary antibody used.

We thank the reviewer for bringing this to our attention. We revised the methods section under “Western Blots” to include this information in our revised manuscript.

4. A short description of the ELISA protocol may be provided.

We thank the reviewer for making this suggestions. We revised the methods section under “ELISA” to include this information in our revised manuscript.

5. The make of the restriction enzymes used may be provided

We thank the reviewer for making this request. We revised the methods section under “Construction of AAV9-sRAGE” to include this information in our revised manuscript.

6. Catalog numbers and brands of all chemicals/kits/buffers may be provided

We thank the reviewer for making this request. We revised the methods section under “Construction of AAV9-sRAGE” to include this information in our revised manuscript.

---

## [Decision Letter · Decision Letter 1]

22 Jan 2025

Soluble RAGE enhances muscle regeneration after cryoinjury in aged and diseased mice

PONE-D-24-50348R1

Dear Dr. Horwitz,

We’re pleased to inform you that your manuscript has been judged scientifically suitable for publication and will be formally accepted for publication once it meets all outstanding technical requirements.

Kind regards,

Ram Nagaraj

Academic Editor

PLOS ONE

Additional Editor Comments (optional):

Reviewers' comments:

Reviewer's Responses to Questions

**Comments to the Author**

1. If the authors have adequately addressed your comments raised in a previous round of review and you feel that this manuscript is now acceptable for publication, you may indicate that here to bypass the “Comments to the Author” section, enter your conflict of interest statement in the “Confidential to Editor” section, and submit your "Accept" recommendation.

Reviewer #1: All comments have been addressed

2. Is the manuscript technically sound, and do the data support the conclusions?

Reviewer #1: Yes

3. Has the statistical analysis been performed appropriately and rigorously? 

Reviewer #1: Yes

4. Have the authors made all data underlying the findings in their manuscript fully available?

Reviewer #1: Yes

5. Is the manuscript presented in an intelligible fashion and written in standard English?

Reviewer #1: Yes

6. Review Comments to the Author

Reviewer #1: The authors have addressed all of the reviewer’s questions, and this paper is now suitable for publication.

7. PLOS authors have the option to publish the peer review history of their article (what does this mean? ). If published, this will include your full peer review and any attached files.

**Do you want your identity to be public for this peer review?** For information about this choice, including consent withdrawal, please see our Privacy Policy .

Reviewer #1: No

---

## [Editor Report · Acceptance letter]

PONE-D-24-50348R1

PLOS ONE

Dear Dr. Horwitz,

I'm pleased to inform you that your manuscript has been deemed suitable for publication in PLOS ONE. Congratulations! Your manuscript is now being handed over to our production team.

Kind regards,

on behalf of

Dr. Ram Nagaraj

Academic Editor

PLOS ONE